# Let-7 as a Promising Target in Aging and Aging-Related Diseases: A Promise or a Pledge

**DOI:** 10.3390/biom12081070

**Published:** 2022-08-02

**Authors:** Ya Wang, Juanjuan Zhao, Shipeng Chen, Dongmei Li, Jing Yang, Xu Zhao, Ming Qin, Mengmeng Guo, Chao Chen, Zhixu He, Ya Zhou, Lin Xu

**Affiliations:** 1Special Key Laboratory of Gene Detection and Therapy & Base for Talents in Biotherapy of Guizhou Province, Zunyi 563000, China; wyy0104@126.com (Y.W.); jj.z.2008@163.com (J.Z.); c2475661482@163.com (S.C.); lidongmei@zmu.edu.cn (D.L.); yangjingwhh@163.com (J.Y.); zhaoxu07122021@163.com (X.Z.); qinming888@126.com (M.Q.); guomengmeng151@163.com (M.G.); cchao0704@163.com (C.C.); 2Department of Immunology, Zunyi Medical University, Zunyi 563000, China; 3The Collaborative Innovation Center of Tissue Damage Repair and Regeneration Medicine of Zunyi Medical University, Zunyi 563000, China; hzx@gmc.edu.cn; 4Department of Medical Physics, Zunyi Medical University, Zunyi 563000, China

**Keywords:** miRNA, let-7, regulation, aging, aging-related diseases

## Abstract

The abnormal regulation and expression of microRNA (miRNA) are closely related to the aging process and the occurrence and development of aging-related diseases. Lethal-7 (let-7) was discovered in *Caenorhabditis elegans* (*C. elegans*) and plays an important role in development by regulating cell fate regulators. Accumulating evidence has shown that let-7 is elevated in aging tissues and participates in multiple pathways that regulate the aging process, including affecting tissue stem cell function, body metabolism, and various aging-related diseases (ARDs). Moreover, recent studies have found that let-7 plays an important role in the senescence of B cells, suggesting that let-7 may also participate in the aging process by regulating immune function. Therefore, these studies show the diversity and complexity of let-7 expression and regulatory functions during aging. In this review, we provide a detailed overview of let-7 expression regulation as well as its role in different tissue aging and aging-related diseases, which may provide new ideas for enriching the complex expression regulation mechanism and pathobiological function of let-7 in aging and related diseases and ultimately provide help for the development of new therapeutic strategies.

## 1. Introduction

Aging is a complex systemic physiological process characterized by progressive deterioration of tissue and organ functions and reduced repair capacity, leading to increased susceptibility to aging-related diseases (ARDs) and increased risk of death. The aging process is associated with various physiological and pathological molecular mechanisms, many of which are differentially expressed by age-dependent transcriptional regulation, resulting in changes in multiple cells, tissues and organs, and even lifespan [1,2]. In addition, aging is a driver of a variety of age-related diseases, including neurodegenerative diseases, age-related cataracts, osteoporosis, etc. [3,4,5]. Although many interventions have been used clinically to slow the progression of aging-related diseases, their effectiveness is still limited [6,7]. Therefore, a better understanding of the molecular mechanisms of the aging process and identification of aging biomarkers and possible therapeutic targets for ARDs is performed to pave the way for future research on aging biology and new antiaging therapies.

Let-7 was originally identified as a heterochronic gene in *Caenorhabditis elegans* (*C. elegans*) and also named lethal-7 (let-7) because its deletion is lethal during development [8]. Let-7 is about 22 nt in length and is highly conserved across animal species [8]. Although the let-7 sequences from nematodes to humans are very conservative, there are great differences in transcriptional regulation among members of the let-7 family due to different genomic sites [9,10]. Currently, 12 genetic loci have been identified as the origin of let-7 in humans [10], 3 in mouse [11], and 1 in drosophila [12]. In humans and mice, let-7 has 10 miRNAs (let-7a, b, c, d, e, f, g, i, and miRNA-98 and miRNA-202) [10]. With the exception of *let**-**7i* and *let**-**7g*, which are encoded individually, transcripts of other members are located in a common gene cluster [13,14,15,16]. In mammals, let-7 has been successively found to regulate cell differentiation, development, and apoptosis and to be involved in glucose metabolism (Figure 1) [8,17,18,19]. It also acts as a tumor suppressor to regulate cell proliferation, and its dysregulation and expression are associated with disease progression [20,21]. Recent studies have revealed that let-7 not only increases in aging tissues and acts as an important regulator of cell and tissue senescence in aging organisms [22] but also plays a key role in the development of aging-related diseases represented by Alzheimer’s disease [23,24]. These studies further suggest that let-7 is a potential intervention target molecule in the pathological process of aging. Therefore, in this review, we focus on the association between let-7 expression and aging in multiple tissues, organs, and aging-related diseases.

## 2. Let-7 Expression and Regulation Mechanism

Let-7 expression is lower in undifferentiated cells and gradually increases within cell differentiation during development [25]. Let-7 first appears shortly after fertilization and is responsible for mediating the attachment of blastocysts to the uterine spiral arteries. In early gestation, let-7 is inhibited by Lin28 in trophoblast cells to promote successful placental development. After full term, let-7 expression is elevated and accompanied by diminished inhibition [26]. Three months later, let-7 is essential for the differentiation of different organs by preventing unwanted cell proliferation, and its elevated expression acts as a “stop” marker in many tissues, including lung [27], brain [28], heart, and vascular tissues [29]. It is suggested that let-7 targets proliferation-related genes in a silencing type to reduce cell proliferation, induce cell differentiation, and persist throughout the later stages of embryonic development from the very beginning [28]. After birth, let-7a or let-7b is minimally expressed in the hypothalamus and subsequently increases gradually and affects growth and puberty in a sex-specific manner [30]. 

Despite multiple loci of origin, all let-7 members begin with the pre-let-7 transcript [31]. Generally, the maturation of let-7 follows the typical miRNA biogenesis pathway, including the primary miRNA (pri-miRNA) produced by RNA polymerase II being processed in the nucleus by the RNase III enzyme Drosha protein and the double-stranded RNA-binding protein DGCR8 to produce precursor miRNA (pre-miRNA). Generated pre-miRNA is dependent on EXP5-Ran-GTP for transport to the cytoplasm and further processed by Dicer to generate mature miRNA that exerts a function in gene silencing (Figure 2a) [31,32,33]. Notably, some family members require an extra step: three members of the let-7 family (let-7a-2, -7c, and -7e) carry typical 2 nt 3′ overhangs in their precursors, while the remaining members only have 1 nt 3′ overhangs [34] because the remaining member pri-let-7 precursor has a bulging adenosine (pri-let-7d) or uridine (all other members) next to the processing site [34]. This uridine/adenosine bulge results in a single nucleotide 3′ overhang. Due to this structural difference, terminal uridyltransferases (TUT2/PAPD4/GLD2, TUT4/ZCCHC11, and TUT7/ZCCHC6) are required to specifically mono-uridylate the 3′ terminus of pre-let-7s, resulting in a 2 nt 3′ overhang which is preferentially bound and cleaved by Dicer (Figure 2b) [34].

The biogenesis of let-7 is strictly regulated by multiple levels of factors. First of all, it has been reported that the let-7 promoter can be activated by OCT-4, a key regulator that maintains the pluripotency and self-renewal characteristics of embryonic stem cells and is inhibited by a variety of proto-oncogenes, such as p53 [35,36]. Secondly, single-nucleotide polymorphisms (SNPs) in the let-7i promoter region can also affect the binding of related transcription factors and thus cause altered expression [37]. Furthermore, in epigenetics, promoter methylation and histone modification can affect the transcription of let-7 [38,39,40]. Recent studies have shown that METTL1 (methyltransferase-like 1)-mediated methylation enhances let-7 processing by disrupting repressive secondary structures within pri-let-7 [41]. It is worth noting that several transcription factors have been found to act as regulators of let-7 biogenesis and are also targeted by let-7. For example, there is a feedback loop between let-7 and the nuclear hormone receptor DAF-12 in *C. elegans* because DAF-12 is a target of let-7 but also regulates the transcription of let-7 in a ligand-dependent manner [42]. A similar phenomenon exists between MYC and let-7. The expression of MYC is repressed by let-7d, but during MYC-mediated tumorigenesis, it can inhibit the transcription of let-7 by directly binding to the promoter and upstream regions of let-7a-1/let-7f-1/let-7d [43]. 

Post-transcriptional regulation of let-7 is carried out on Lin28A and Lin28B. The histone H3K4 methyltransferase SET7/9 can monomethylate the lysine 135 of Lin28A, which binds to pri-let-7 in the nucleus and sequesters it into the nucleolus to prevent Drosha mediated processing [44]. Lin28B has nucleolar localization sequence (NoLS), so can be located in the nucleolus. Seemingly, Lin28B binds to pri-let-7 in the nucleus directly and sequesters it to Drosha-deficient nucleoli, thereby inhibiting let-7 maturation through a TUTase-independent pathway (Figure 2d) [45]. Interestingly, a recent study showed that Lin28B interacted with DIS3L2 in the cytoplasm of Lin28B-expressing cancer cell lines, suggesting that it was also involved in a TUTase-dependent pathway [46]. In this case, the level of pre-let-7 appears to affect the subcellular localization of Lin28B [46]. In addition, it was revealed that in Lin28A overexpressed in human embryonic stem cells or cancer cells, the 3′-terminus of pre-let-7 is oligonucleotidylated by TUT4 and TUT7 to produce a uridine tail of approximately 14 nt, which resists Dicer cleavage but is readily degraded by the cytoplasmic exosome DIS3L2 catalytic (Figure 2c) [47,48,49,50,51]. However, what happens when pri-let-7 is sequestered into the nucleolus by methylated Lin28A or Lin28B is puzzling, as is what the specific mechanism is by which Lin28 acts on the 3′-terminus of pre-let-7 to initiate oligouridylation; therefore, the details of the relationship between DIS3L2-associated cytoplasmic exosomes and let-7 biogenesis are also unknown (Table 1).

## 3. Let-7 and Lifespan

Many miRNAs have been shown to directly affect lifespan through insulin-like growth factor (IGF-1) signaling, target of rapamycin (TOR), and mitochondrial/reactive oxygen (ROS) signaling, among other pathways [75]. It is known that let-7 downregulates insulin-PI3K-mTOR signaling in mammals [19], and insulin-PI3K-mTOR signaling promotes aging in an evolutionarily conserved manner, indicating let-7 slows aging in mammals. However, it has been reported that upregulation of miR-48 and miR-84 (let-7 family members) after TDCPP exposure in *Cryptobacterium hidradenum* ultimately contributes to the reduction of nematode lifespan and locomotor behavior by silencing DAF-16/FoxO in the unconventional insulin-like growth factor signaling (IIS) pathway [76]. Moreover, other substantial evidence shows that let-7 affects lifespan through other different biological pathways. For example, Xu et al. showed that the expression of adaptor proteins p66Shc, p52Shc, and p46Shc were regulated by let-7a at the post-transcriptional level [77]. Among them, the p66Shc adaptor protein was considered to be a key regulator of mammalian lifespan, and the inhibition of p66Shc expression by let-7a delayed the senescence of human diploid fibroblasts (HDF). Gendron et al. further found that in *Drosophila melanogaster*, let-7 overexpression in neurons specifically elevated female triglycerides (TAG), resulting in a significant increase in female mean lifespan (22%) and maximum lifespan (14%), which was determined to extend lifespan in a manner independent of changes in nutritional intake or systemic insulin signaling [78]. What cannot be ignored is that the functions of let-7 are complex and diverse, such as let-7 overexpression in male *Drosophila* neurons significantly shortening lifespan [78], while previous work has shown that maintaining let-7c expression is necessary for lifespan in healthy adult male *Drosophila* [79], and the exact reason for the gender dimorphism of let-7 effects on lifespan is also unknown. Meanwhile, it is worth mentioning that exercise training appears to promote healthy biological aging by inducing telomere maintenance, but the molecular mechanisms are not fully understood. Kumard et al. reported that let-7 and miR-320 were co-expressed and downregulated after short interval training (SIT), suggesting that altered expression of let-7 and miR-320 is associated with delayed biological aging after SIT [80].

In addition, excessive tissue regeneration may shorten lifespan by leading to stem cell depletion or tumorigenesis [81,82]. Remarkably, let-7 depletion promotes cardiac regeneration [83] and liver regeneration [82]. Therefore, it is important to accurately grasp the pleiotropic nature of let-7 and reasonably implement targeted interventions in order to achieve an optimal balance of tissue regeneration and ultimately prolong lifespan.

### 3.1. Let-7—The Role in Tissue Aging and ARDs

#### 3.1.1. Let-7 and the Nervous System

Let-7 is widely expressed in the aging central nervous system, resulting in decreased neuron formation and the self-renewal of neural stem cells. Hmga2, highly expressed in fetal and young animal neural stem cells (NSCs), promotes self-renewal of neural stem cells at least in part through downregulation of p16 ^Ink4a^ and p19 ^Arf^ [84]. Studies have shown that Hgma2 mRNA contains seven let-7 binding sites, for example, let-7b binding to Hmga2 3′UTR negatively regulates its expression [85]. Thus, due to increased let-7 levels in NSCs of aged animals, reduced Hmga2 expression leads to elevated cell cycle inhibitors [79], thereby halting NSC renewal [24]. In addition, in older neurons, let-7 upregulation inhibits the expression of Lin-41, an important promoter of anterior ventral microtubule (AVM) axon regeneration [86], which in turn inhibits AVM axon regeneration and ultimately leads to a decline in neuronal regeneration capacity. Additionally, the binding of let-7 and Lin-28 controls the maintenance of DA9 synaptic polarity [87], which is important in aging and neurodegenerative diseases. 

Alzheimer’s disease (AD) is a devastating neurodegenerative disease caused by the accumulation of amyloid plaques and hyperphosphorylated tau in the brain [24], and multiple miRNAs are involved in this critical pathological process [88]. The major component of these plaques, β-amyloid peptide (Aβ), is derived from the amyloid precursor protein (APP) through the sequential action of β- and γ-secretase [24]. *Drosophila melanogaster* and *C. elegans* do not have the APP gene, but both express the amyloid precursor protein gene APL-1 [89], which is controlled by let-7 and its targets. For instance, let-7 transcriptionally regulates APL-1 through Hbl1, Lin-41, and Lin-42 and is critical for the development of AD [90]. Moreover, it is also known that let-7 is negatively regulated by miR-107. Interestingly, studies have shown that miR-107 downregulated beta amyloid cleaving enzyme (BACE1), which generated amyloid beta peptide (Aβ) fragments by cleaving APP, leading to amyloid beta deposition promoting the development of AD [91]. However, whether the apparent down-regulation of miR-107 in AD [92] negatively up-regulates let-7 expression [93] and promotes the development of AD needs to be further verified. In addition, dysregulation of miRNAs leads to dysfunction of intracellular and extracellular biochemical processes and ultimately to neuronal cell death, which is another factor influencing AD [94,95]. For example, the expression of let-7a, let-7b and let-7e are found to be upregulated in high cholesterol diet–induced AD progression in a late-onset rabbit mode [96], suggesting that let-7 may be involved in the pathological process of cholesterol metabolism associated with AD. Another study reported that the cerebrospinal fluid of AD individuals was specifically enriched in let-7b or let-7e [97] and interacted with TLR7 receptors as signaling molecules, further activating IRAK-4 through phosphorylation, and the activated IRAK-4 stimulates caspase-3 to activate the TLR7 signaling pathway, which eventually leads to neuronal degeneration [98]. Recent studies suggest that abnormal autophagy may be a major risk factor for AD [99], as shown by Gu et al., who revealed that let-7a overexpression in concert with the PI3K/AKT/mTOR signaling pathway enhances Aβ1-40-induced neurotoxicity through the regulation of autophagy [100].

Parkinson’s disease (PD) is the second most common age-related disease and is caused by a region-selective loss of dopaminergic neurons in the substantia nigra pars compacta [24]. PD is characterized by accumulation of Lewy bodies in dopaminergic neurons due to mutations in α-synuclein, Perkin, UCHL1, DJ1, and LRRK2 genes [101,102]. Recently, it has been reported that let-7 downregulates E2F1, a conserved cell cycle-related transcription factor that activates cell death through multiple pathways, and mutant leucine-rich repeat kinase 2 (LRRK2) counteracts let-7, resulting in excessive expression of E2F1 and eventually causing dopaminergic neuron death [103]. In addition, deletion of let-7 leads to changes in various PD-related pathways, such as decreased α-synuclein expression and accumulation, increased autophagy, and increased oxidative stress [104]. This evidence shows that let-7 is an important regulator of the PD process and may be a new intervention target for the treatment of PD.

The senescence of hippocampal neural stem cells (H-NSCs) can lead to cell exhaustion, neurogenesis reduction, and cognitive impairment in vascular dementia (VD) [105]. Several miRNAs, including let-7a-5p in embryonic stem cell-derived small extracellular vesicles (ESC-sEVs), ameliorate H-NSCs senescence by inhibiting mTORC1 activation, promoting TFEB nuclear translocation and lysosomal recovery, thereby reversing the neurogenetic and cognitive disorders associated with senescence in VD [105]. 

Taken together, this evidence shows that let-7 is involved in multiple biological processes associated with neurological aging and closely associated with the development of aging-related neurological disorders. Importantly, existing studies reveal that let-7 not only serves as a biomarker in neurological aging and related diseases but also can reverse aging-related neurological impairment and cognitive deficits, indicating it is an important therapeutic target for aging-related neurodegenerative diseases.

#### 3.1.2. Let-7 and the Vision System

Let-7 is involved in the regulation of retinal development and the cell cycle, and its expression gradually increases with age [106]. For example, let-7b and let-7c increase significantly during normal vitreous aging, and both of them are expressed by Muller glia cells and detected in their extracellular vesicles [106]. Further studies have found that let-7c targets hyaluronic acid synthase 2 (Has2), a major component of vitreous synthase, which affects vitreous development and remodels during aging by regulating hyaluronic acid content with binding to the 3′UTR sequence, promoting the structural changes of the extracellular matrix [106]. Studies have shown that let-7 is associated with several age-related retinal diseases. For example, dysregulation of let-7 family members has been detected in the vitreous fluid of both age-related macular degeneration and proliferative retinopathy patients [23,107]. However, the specific role of let-7 in aging-related eye diseases remains to be elucidated.

Aging is the main cause of cataracts [108]. Some findings suggest that miRNAs play a role in age-related cataracts [109]. Let-7b, one of the top eight miRNAs in the human transparent and age-related cataract lens microarray [110], regulates UV-induced lens epithelial cell apoptosis by directly targeting G protein-coupled receptor 4 (Lgr4) [111]. In addition, there is a close relationship between oxidative stress and age-related cataracts [112]. It is found that let-7c-3p is downregulated in the anterior capsule of the lens in the age-related cataracts group aged >65 years compared to the age-related cataracts group aged ≤65 years. Similarly, the expression of let-7c-3p is lower under oxidative stress [108]. Recently, let-7c-3p was found to inhibit oxidative stress-induced apoptosis in lens epithelial cells (LECs) and target ATG3 to reduce autophagy levels [113]. This implies that let-7c-3p may be a new target for age-related cataracts treatment.

#### 3.1.3. Let-7 and the Reproductive System

The trade-off between the allocation of resources and energy for reproduction and growth versus somatic maintenance has been central to the “evolutionary optimization” theory of aging [114], and the removal of germline stem cells by germline laser ablation or GLP-1/NOTCH mutants can indeed extend the lifespan of nematodes [115]. A study found that knocking out Lin28 extends lifespan and promotes germline stem cells into meiosis in nematodes, resulting in far fewer germline stem cells in young adults [116]. Importantly, as the best-known downstream effector of Lin28, let-7 stimulates DAF-16 translocation by targeting AKT-1/2, which is essential for Lin28-induced longevity and smaller germline progenitor cell pools, and the reproductive stem cell and lifespan effects of Lin28 RNAi are eliminated in let-7, AKT-1/2, and DAF-16 mutant worms, indicating that the Lin28/let-7/AKT/DAF-16 axis plays an important role in balancing reproduction and somatic cell maintenance [117]. Furthermore, Lin28 is specifically expressed in the niche of testicular stem cells, directly binding and protecting Upd mRNA (a stem cell self-renewal factor), maintaining the number and function of central cells in the testicular stem cell ecotone [118]. However, Lin28 expression decreases with age, causing let-7 to bind IGF-II messenger RNA by targeting protein (Imp) to reduce Upd expression, leading to a consequent loss of germline stem cells [119]. Furthermore, a downregulation of let-7c is detected in patients with premature ovarian failure (POF) compared to normal women [120], implying an active role of let-7c in healthy follicle development. However, the function of let-7g in follicles seems to be different from other family members because it is highly expressed during atresia [121], directly targets the anti-apoptotic gene MAP3K1, and causes the expression and dephosphorylation of the transcription factor FoxO1, which in turn induces GC (granulosa cell) apoptosis [122]. Meanwhile, let-7g targets TGFBR1 to block the TGFβ signaling pathway and increase caspase-3 activity and the apoptosis rate [123]. Fortunately, in a typical mouse model of premature ovarian failure (POF) induced by a high-fat, high-sugar (HFHS) diet, thymopentin promotes the transcriptional activation of Lin28A by stimulating the expression of transcription factor YY2, inhibiting the activity of let-7 family miRNAs and alleviating the senescence of ovarian granulosa cells, ultimately achieving the therapeutic effect on POF in mice [124]. 

Therefore, the search for more upstream transcriptional regulators of Lin28 targeting let-7 has positive clinical therapeutic value for maintaining germinal stem cell populations, delaying germ cell senescence, and aging-related reproductive disorders.

#### 3.1.4. Let-7 and the Immune System

Aging is accompanied by adaptive immune system senescence including the dysfunction of B and T lymphocytes [125], and miRNAs have a key regulatory role in immune cell development and function [126,127]. There is growing evidence to support the role of let-7 in the regulation of the immune system, including let-7 as a key player in the regulation of B-cell antibody production, T-cell activation, and macrophage responses, etc. [128,129,130,131]. However, little is known about the relationship between let-7 and the immune system during aging. A reduction in the size of the pre-B-cell pool in aging mice has been reported [132], accompanied by intrinsic B-cell defects [133], but the underlying causes of these changes remain to be elucidated. Koohy et al. found that compared with young cells, the level of IRS1 protein in aging pro-B cells and pre-B cells decreased, while the expression level of let-7 increased [134]. Importantly, downregulation of Irs1 and upregulation of let-7 expression are major components of the transcriptional downregulation of the insulin-like growth factor signaling pathway in aging, and therefore upregulation of let-7 in the aging B-cell precursors targeting Irs1 and other components at the post-transcriptional level is likely to result in reduced responsiveness to insulin/IGF signaling [134]. Interestingly, there are striking alterations in the chromatin at a novel potential precursor RNA for the let-7b and -7c2 in pre-B cell senescence [134], which is indicative of an interplay between epigenetic and post-transcriptional mechanisms in shaping gene expression. This finding suggests that aging affects the regulation of key signaling pathways at multiple levels. 

Inflammation is one of the markers of accelerated aging and age-related diseases [135,136,137]. Meaningfully, the let-7 family also plays a key role in regulating immune-mediated inflammation [138]. For example, the let-7adf cluster both inhibits Tet2 expression and increases succinate accumulation by regulating the Lin28A/Sdha axis in LPS-activated macrophages to enhance IL-6 secretion enzymes in macrophages [130]. However, whether let-7 regulation of tissue inflammation is involved in the process of aging and the associated molecular mechanisms remains unknown. In addition, current studies mostly focus on the potential of let-7 as a target for cancer immunotherapy. Therefore, an in-depth understanding of the relationship between let-7 and immune cell subsets during the aging process may help provide new immunological treatment strategies for the intervention of aging.

#### 3.1.5. Let-7 and Other Organizations

Muscle loss is a main contributor to aging-related diseases, and strategies to improve muscle regeneration during aging are urgently needed. The role of individual miRNA in skeletal muscle formation has been extensively studied, and many miRNAs have been shown to be involved in this process [139]. As reported in the study, let-7b and let-7e are significantly elevated in skeletal muscle in the elderly and cause skeletal muscle loss by targeting cell cycle regulators CDK6, CDC25A, and CDC34 [140]. Other studies have shown that single inhibition of let-7 promotes muscle cell differentiation increased muscle mass in mice [19,141,142]. Surprisingly, inhibition of a combination of five miRNAs containing let-7 increased activation of focal adhesion kinase (FAK), AKT, and p38 mitogen-activated protein kinase (MAPK) during myogenic differentiation and improved myotube formation and insulin-dependent glycogen synthesis [143]. In addition, let-7-targeting paired cassette 7 (PAX7) and IL-6 in senescent muscle and oculopharyngeal muscular dystrophy (OPMD), a disease that shares molecular characteristics with senescent muscle, leads to reduced muscle regeneration and functional degeneration [144]. 

Osteogenesis and adipogenesis of bone marrow mesenchymal stem cells (MSC) maintain homeostasis in vivo under physiological conditions. With the increase of age, the balance between adipogenesis and osteogenic differentiation of MSCs may be disrupted, resulting in excessive accumulation of bone marrow adipocytes and reduction of bone mass, which is related to age-related bone metabolic diseases (such as osteoporosis) [145]. Let-7 is found to positively regulate osteogenic differentiation and negatively regulate lipogenic differentiation of HADSCs by inhibiting Hmga2 while negatively regulating lipogenic differentiation, suggesting that let-7 is therefore a positive regulator of skeletal development [146]. Importantly, let-7c and let-7d expression in mouse femurs increased after birth and peaked at 4 weeks, followed by a rapid decline after bone maturation, indicating that let-7 expression coincides with the timing of skeletal development. Thus, downregulation of let-7 expression in bone with increasing age reduces osteogenesis by inhibiting osteogenic differentiation of HADSC. Notably, another study found that let-7g expression is downregulated in osteoporosis, but let-7g mimics inhibit ALP activity and mineral deposition in osteoblasts, which in turn controls osteoblast differentiation [147]. Conversely, let-7c is highly expressed in the serum of postmenopausal patients with osteoporosis and, by targeting SCD-1, shuts down Wnt/β-catenin signaling and ultimately inhibits osteogenic differentiation [148].

The above findings suggest that let-7 family members affect age-related bone production in different ways and may also be promising drug targets for bone loss–related diseases.

## 4. Conclusions and Future Directions

Collectively, let-7 is highly expressed in multiple tissues, including brain [24,98,103], retina [106,111], and muscle [139,141], which regulates the differentiation function of multi-tissue stem cells by targeting different genes (Figure 3) to regulate aging-related pathways, thereby affecting the development of aging and related diseases, suggesting that let-7 plays critical role in the process of aging and ARDs and is a promising target for intervention.

However, there are still some essential scientific issues remaining to be fully elucidated, which might be the critical directions of the application of let-7-based intervention strategies (Figure 4). First, the underlying mechanisms among these members of let-7 family in different biological process remain to be fully elucidated, which is essential for the development of let-7 family-based therapeutic targets. Evidence suggests that different members of the let-7 family have different functions in specific cell types. For instance, let-7a, -7b, and -7c are highly expressed in neurons and contribute to the development of AD through different mechanisms, but whether there is network synergy among the three is unknown. Moreover, the mechanisms of aging are complex, and metabolic pathways [149], immune function [150], and telomere length [151] are all key factors in the aging process, while, the role of let-7 family in these processes has not been fully elucidated. For example, T2DM is a potential risk factor for aging-related diseases [152], and the increased expression of let-7a and let-7d in the skeletal muscle of patients with T2DM directly inhibits the translation of IL-13 mRNA and reduces glucose uptake and metabolism [153]. So, does let-7-induced impaired glucose and lipid metabolism lead to decreased muscle mass and increased risk of aging-related diseases? Additionally, what are the functional effects and mechanisms of let-7 on immune cell subsets during aging? Whether let-7 is involved in telomere length regulation after intermittent exercise requires further in-depth exploration to elucidate these questions. Second, the expression pattern and profile of the let-7 family in the processes of different biological events remains to be further verified, which is important for the development of let-7-based biomarkers for these events. For example, in hepatocellular carcinoma, due to intrinsic different expression levels of distinct let-7 family members, overexpression of different let-7 family members has been shown to have different degrees of effect on inhibiting cell viability, in which let-7a has the greatest effect [154]. However, the distinct pattern of let-7a during the carcinogenesis remains to be explored. Meanwhile, the specific impact of abnormal let-7 family members in the vitreous humor on several aging-related ocular diseases is also unclear, so it is inconclusive whether it can be used as a differential diagnostic marker for them. Third, in the aspect of clinical application, although targeting miRNA is a promising treatment strategy for aging, a fully realized treatment may require long-term efforts. Clinical data show that let-7 can be used as a biomarker for cancer screening and diagnosis and is a promising target for cancer therapy [155]. However, aging-related let-7 therapy has not been established because of some unresolved issues, including in the design of miRNA therapy for aging and age-related pathology, enhancing targeting [156] and delivery efficiency [157], as well as safety [158], which are still the directions of efforts.

In all, accumulating evidence has revealed that let-7, as an intrinsic regulator, plays critical roles in the development of aging and aging-related diseases. Further clarifications of the expression regulation mechanism and various pathological functions of let-7, as well as its value in clinical diagnosis and in aging and related diseases, are highly valuable for the illustration of let-7 biological function in aging and aging-related diseases and ultimately benefit the development of new therapeutic strategies in the future.

## Figures and Tables

**Figure 1 biomolecules-12-01070-f001:**
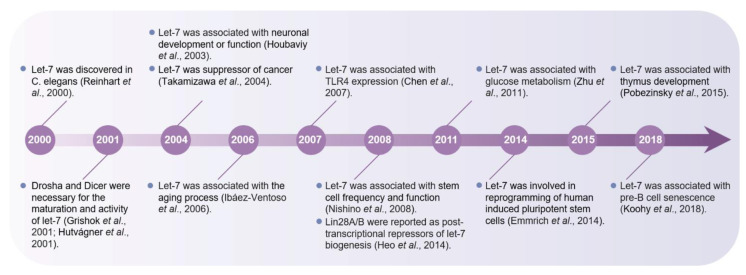
Timeline of major research discoveries related to let-7.

**Figure 2 biomolecules-12-01070-f002:**
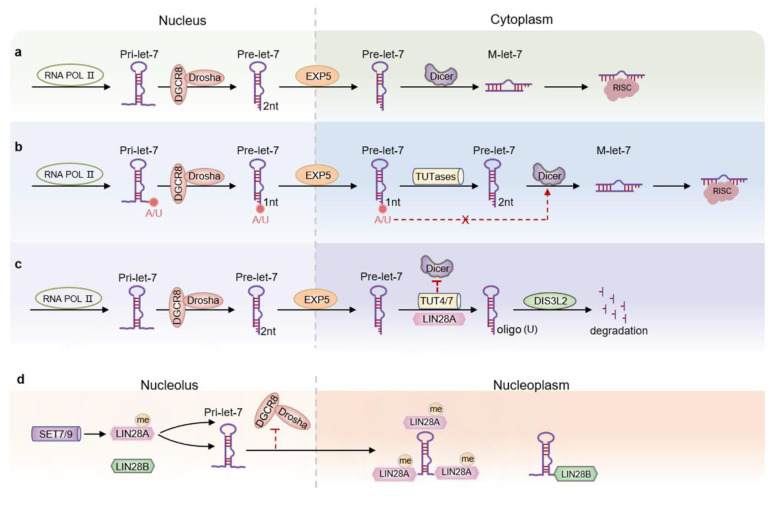
Biogenesis and regulatory pathways of let-7 family members. (**a**) Let-7 follows the typical miRNA biogenesis pathway. (**b**) TUTases specifically mono-uridylate the 3′ end of the 1 nt, yielding the 2 nt 3′ overhang preferred by Dicer to facilitate processing of pre-let-7. (**c**) Pre-let-7 is oligonucleotidylated at the 3′ end by Lin28A and TUT4/7 and is resistant to cleavage by Dicer but sensitive to catalytic degradation by DIS3L2. (**d**) The methylated Lin28A and Lin28B nucleus bind to pri-let-7 in the nucleus and segregate it into nucleosomes, preventing Drosha-mediated processing.

**Figure 3 biomolecules-12-01070-f003:**
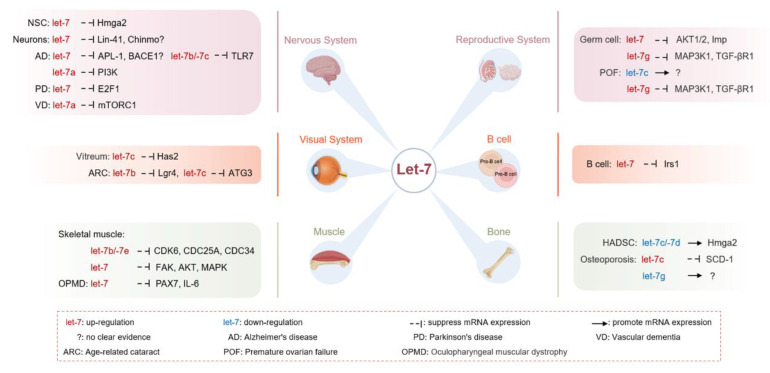
Target genes regulated by let-7 in aging and aging-related diseases in multiple tissue systems.

**Figure 4 biomolecules-12-01070-f004:**
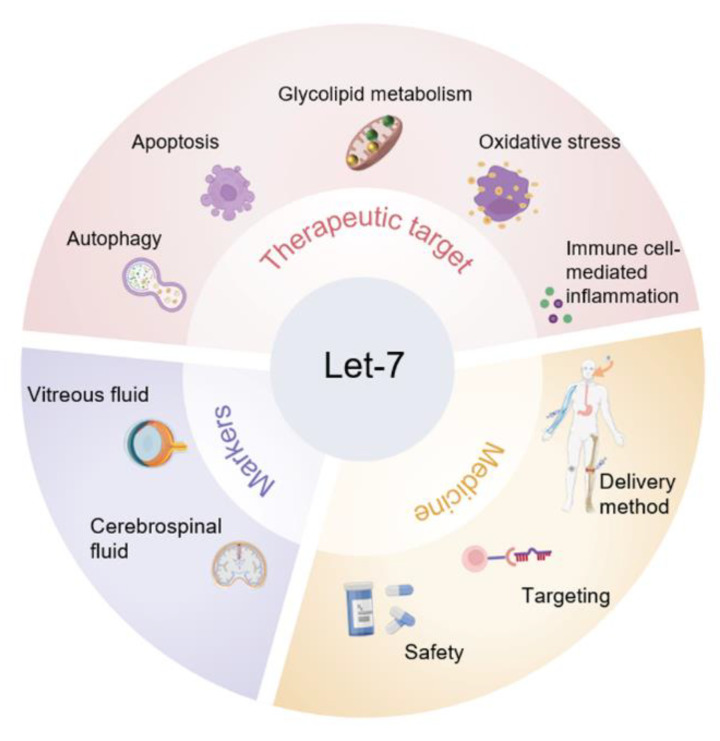
Future research directions of let-7 in aging-related diseases.

**Table 1 biomolecules-12-01070-t001:** Some known regulatory proteins that affect let-7 biogenesis through different pathways.

**Inhibitory Regulatory Protein**	**Family Member**	**Mechanism**	**Ref.**
Lin42	let-7a, 7b	Suppresses let-7 transcriptionally by binding to the pri-let-7 3′UTR	[8,52]
Lin28A-TUTases4/7	let-7a, 7b, 7d, 7g, 7i	Represses let-7 through TUTase-dependent uridylation of pre-let-7	[48,49]
Lin28B	let-7a, 7d, 7f, 7g, 7i	Represses let-7 by sequestering pri-let-7 into the nucleolus	[45,53]
TRIM25	let-7a	Activates TuT4, allowing for more efficient Lin28A-mediated uridylation	[54]
MUC1-C	let-7c	Activates Lin28B and synergistically represses let-7	[55]
MSI1	let-7b, 7g, miR-98	Recruits Lin28 to the nucleus and represses let-7	[56]
FHIT	let-7a, 7b, 7d, 7f, 7g	Induces Lin28B protein, consequently inhibiting let-7	[57]
NF90/NF45	let-7a	Directly binds to pri-let-7 and interacts with Drosha complex to inhibit pri-let-7 processing	[58]
YAP	Let-7g	Translocates into the nucleus and sequesters DDX17 and interferes with Drosha processing	[59]
hnRNPA1	let-7a	Reduces Drosha processing	[60]
TRAIL-R2	let-7a, 7b, 7c, 7d, 7e, 7g	Interacts with Drosha and DGCR8 to inhibit pri-let-7 processing	[61]
MCPIP1	let-7g	Cleaves terminal loops on the pre-let-7 leading to degradation	[62]
STAUFEN	let-7s	Likely binds to pri-let-7 3′UTR and negatively modulates let-7	[63]
SSB	let-7a, 7b, 7c, 7d, 7e, 7f, 7g, 7i	Positively regulates Lin28 to suppress the maturation of let-7	[64]
**Activating Regulatory Protein**	**Family Member**	**Mechanism**	**Ref.**
METTL1	let-7e	METTL1-mediated methylation augments let-7 processing by disrupting an inhibitory secondary structure within the pri-let-7 transcript	[41]
TUTases2/4/7	let-7a, 7b, 7d, 7f, 7g, 7i, miR-98	Specific mono-uridylation of pre-let-7 for preferential binding and cleavage by Dicer	[34]
SNIP1	let-7i	Likely binds pri-let-7 and enhances Drosha processing	[65]
TTP	Let-7a, 7b, 7f, 7g	Enhances let-7 expression by down-regulation of Lin28A expression	[66]
KSRP	let-7a	Promotes let-7 maturation as part of Drosha and Dicer complexes	[60]
RBM3	let-7a, 7g, 7i	Binds pre-let-7s/enhance Dicer	[67]
BRCA1	let-7a	Enhances pri-let-7s processing mediated by Drosha complex	[68]
TDP-43	let-7b	Promotes microRNA biogenesis as a component of the Drosha and Dicer complexes	[69]
TRIM71	let-7a, 7b, 7c, 7d, 7e, 7f, 7g, 7i, miR-98	Negatively regulates Lin28B through polyubiquitination	[70]
BCDIN3D	let-7b, 7d, 7e, 7f, 7g, 7i, miR-98	Methylates pre-let-7s and enhances Dicer processing	[71]
SYNCRIP	let-7a	Binds to pri-let-7 terminal loop and enhances Drosha processing	[72]
**Inhibitory/Activating Regulatory Protein**	**Family Member**	**Mechanism**	**Ref.**
DAF-12	let-7 family	Unliganded DAF-12 represses let-7 and liganded DAF-12 promotes let-7 transcriptionally through binding to pri-let-7 3′-UTR	[42]
MYC	let-7a, 7d, 7f	Inhibited let-7 promoter activity via binding to the noncanonical E-box 3 downstream of the transcription start sites Enhanced promoter activity by binding to the canonical E-box 2 upstream of the transcription start sites	[43]
ADAR1	let-7 family	Directly binds and edits pri-let-7d transcripts thereby reducing the expression of mature let-7d Enhances Drosha and Dicer processing through direct interactions	[73,74]

## Data Availability

Not applicable.

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
