# Peer review of "Let-7 as a Promising Target in Aging and Aging-Related Diseases: A Promise or a Pledge"

_biomolecules, 2022, doi:10.3390/biom12081070_

Round 1

Reviewer 1 Report

I have gone through the review article thoroughly. The author reviewed the recent studies highlighting Let-7 as a promising target in aging and aging-related diseases. The scope of the review is interesting. The article is nicely written and the authors provide nice figures to explain the role of the molecule. I feel this review is an important addition to the literature. This review provide new ideas for enriching the complex expression regulation mechanism and pathobiological function of let-7 in aging and related diseases. This will ultimately pro- 24 vide help for the development of new therapeutic strategies.

I recommend the paper for publication in Biomolecules

Reviewer 2 Report

suggested minor edits:

1. line 42: C. elegans --> Caenorhabditis elegans

2. Line 44: " ,,, across animal species excluding plants." -->

3. Line 145: cryptobacterium hidradenum --> Cryptobacterium hidradenum

                                                                                               Italicize this

4.  Lines 153-158:  Indicate that Gendron et al. were studying Drosophila 

melanogaster.

Conclusion and Future directions.

... and muscle[113,115].  It regulates ... and related diseases.  Let-7 plays a 

critical role in aging and aging-related diseases.

What about future directions?  
